# On the Adaptive Buffer-Aided TDMA Uplink System with AoI-Aware Status-Update Services and Timely Throughput Traffics [note 1]

**DOI:** 10.3390/s24020506

**Published:** 2024-01-13

**Authors:** Tianheng Wang, Qingchun Chen, Shuo Wang, Lei Zheng

**Affiliations:** School of Electronics and Communication Engineering, Guangzhou University, Guangzhou 510006, China; 2112007062@e.gzhu.edu.cn (T.W.); 2112019049@e.gzhu.edu.cn (S.W.); lzheng@gzhu.edu.cn (L.Z.)

**Keywords:** status-update service, throughput-demand service, TDMA uplink, Lyapunov optimization

## Abstract

In this paper, we study a buffer-aided TDMA uplink network, where multiple status-update devices and throughput-demand devices are supposed to upload their data to one information access point (AP), and all devices are assumed to be provisioned with a data buffer to temporarily store the randomly generated data from either the installed sensor or upper-layer applications. To fulfill the communication requirements using two types of devices, the average Age of Information (AoI) is utilized to characterize the data freshness of the status-update devices, while the average sum rate is employed to capture the average transmission performance of the throughput-demand devices. On this basis, a joint-optimization problem was formulated to minimize the average AoI for status-update devices and to maximize the average sum rate for the throughput-demand devices. Lyapunov optimization framework was used to solve the problem of obtaining an AoI-aware adaptive TDMA uplink scheme. Numerical results are presented to show that an AoI-aware adaptive TDMA uplink scheme can effectively fulfill the heterogeneous service requirements using status-update devices and throughput-demand devices.

## 1. Introduction

5G communication technology has made various IoT applications more widespread [1,2,3], and typical IoT applications like augmented reality (AR), autonomous driving, industrial automation, and remote healthcare, have made new requirements on data timeliness. Recently, Age of Information (AoI) was presented as a performance metric to characterize the freshness of information [4,5], and has attracted extensive attention from industry and academia. AoI refers to the time between the present time and the generation time of the latest data packet at the source, which allows the effective measurement of the information timeliness of the received data packets at the destination.

There have been several research studies on AoI-minimization problems in various wireless communication scenarios [6,7,8,9,10,11]. Most of those problems are modeled as a constrained Markov decision process (CMDP). In addition, in many communication scenarios, such as the control of smart-home devices, it is not necessary to have very low latency. Users can tolerate delays of several seconds if the responsiveness of the devices is within a reasonable range. City-level IoT systems (such as traffic monitoring) can tolerate some delay as long as the data are still timely for decision-making and planning. In agricultural IoT applications, such as soil moisture monitoring and irrigation control, the delay in data transmission can be relatively high if it ensures that the crops are still being properly cared for. The authors in [12] investigated multiple-access uplink of wireless communication networks consisting of multiple sensors sharing orthogonal subchannels, and the Lyapunov optimization framework was used to jointly optimize the sampling actions, the allocation of transmit power, and the allocation of subchannels for each sensor subject to long-term average AoI, as well as maximum transmit power constraints. To optimize the transmission of information in wireless communication systems, the problem of maximizing concealment under average AoI constraints has been studied [13], in which covert transmitters need to secretly send information to receivers while creating artificial jamming noises to confuse eavesdroppers. The resource-allocation problem of minimizing long-term power consumption under AoI constraints in a finite block-length (FBL) mechanism for vehicle-to-vehicle (V2V) communication has been investigated [14]. The AoI-aware sensor data collection in mobile edge computing framework has been investigated [15], where wireless and computational resources are jointly allocated to maximize the overall throughput of the system while meeting the maximum AoI threshold constraint. The problem of maximizing long-term average throughput has been addressed [16] to derive a simple static stochastic policy, which does not require any *age-independent* steady knowledge or randomization (AI-SRP). The problem of minimizing weighted-average AoI with timely node throughput constraints has been studied [17] for a single-hop wireless network, where multiple nodes are supposed to transmit time-sensitive information to a base station (BS). It is shown that the maximum weight and drift-plus-penalty strategy using the Lyapunov optimization framework outperforms other strategies in terms of AoI and throughput. In [18], the weighted-average AoI of vehicles at the BS was minimized while maintaining the transmission throughput requirements of vehicles, where the vehicles sensed information about themselves and their surrounding environments through sensors and transmitted them to the BS through uplink channels. A virtual queue method was used to provide dynamic scheduling for the ordered scheduling of multiple vehicles on the road. In [19], the age fairness issue was addressed in vehicle to infrastructure (V2I) networks, where deep reinforcement learning (DQN) training and testing method was employed to learn the dynamic environment and predict the optimal minimum contention window (MCW), such that the optimal MCW can be allocated among different intelligent vehicular nodes.

The coexistence of multiple heterogeneous services imposes challenges for the future development of the IoT, where some emphasize the throughput of data transmission, while others require the freshness of data transmission. In an autonomous driving scenario, the information access point needs to collect real-time location information of the vehicle and provide the driver or passenger with a large amount of data traffic for onboard entertainment. In remote healthcare applications, some medical devices need to provide real-time health data about the patient, while others need to provide surveillance video of the patient. In [20], two types of traffic, namely status-update traffic and timely throughput traffic, were considered. To improve the information freshness of status-update traffic while satisfying timely throughput constraints, an age-aware policy was proposed to ensure the scheduling of decisions is directly based on the current AoI, and an upper bound for the weighted-average AoI was presented under this policy, subject to timely throughput constraints. The authors in [21] studied the average AoI-minimization problem with timely throughput constraints for a wireless network consisting of an AoI-oriented user and a timely throughput user, and the tradeoff between the average AoI and timely throughput was revealed.

A buffer-aided mechanism can better utilize the time diversity gain and efficiently utilize the transmission opportunities to improve transmission throughput [22,23,24,25,26]. In [27,28], the transmission throughput maximization problem for throughput-demand services was studied for a three-node relay network and uplink multiple-access wireless network, respectively, where the average AoI constraints for the status-update services with random arrivals were required. The analysis results show that by explicitly introducing the AoI requirement in adaptive transmission design, we can devise an adaptive transmission scheme to effectively maximize the throughput performance for throughput-demand devices while strictly meeting the AoI requirements using status-update devices. Considering the heterogeneous services in future IoT applications, ensuring different service access requirements is of great importance. To the best of our knowledge, few research studies have considered buffer-aided uplink multiple-access systems with AoI-aware status-update services and timely throughput traffic, and this is exactly the research motivation of our work reported in this paper.

In this paper, we focus on a buffer-aided TDMA uplink wireless communication system with multiple status-update devices and multiple throughput-demand devices. The status-update devices are supposed to report their timely status, and the throughput-demand devices are supposed to deliver as much data as possible. To flexibly support two types of devices, the adaptive TDMA uplink transmission design issue has been addressed [28]. However, only the AoI constraints of all status-update devices were considered in the system design, which did not show the minimum achievable AoI performance in the TDMA uplink system with AoI-aware status-update services and timely throughput traffic. In this paper, we further studied a joint-optimization scheme, where both the throughput performance of all throughput-demand devices and the AoI performance of all status-demand devices are jointly optimized. According to the maximum allowed AoI constraints, two joint-optimized adaptive buffer-aided TDMA uplink schemes, namely *joint optimization with the maximum allowed AoI constraints* (JOWAC) and *joint optimization without the AoI constraints* (JO), are investigated. Our analysis shows that the JOWAC scheme can obtain the best average AoI performance while guaranteeing the strictest average AoI constraints by all status-updated devices. The JO scheme can only realize reasonable average AoI performance when the status-update arrival rate is small enough. The contributions of this paper can be summarized as follows:•This paper considers a buffer-aided TDMA uplink system composed of status-update devices with average AoI requirements and throughput-demand devices with averaged sum-rate performance requirements. A joint-optimization problem was formulated to maximize the average sum rate of the throughput-demand devices and to minimize the average AoI of the status-update devices, subject to the data-queue state evolution constraint and peak transmit power constraint for both types of devices. By solving the joint-optimization problem with the Lyapunov optimization framework, we obtain two AoI-aware adaptive TDMA uplink schemes, namely the JOWAC scheme and the JO scheme.

•Numerical results are presented to highlight that the JOWAC-based AoI-aware adaptive TDMA uplink scheme provides us with a feasible scheme to flexibly fulfill heterogeneous service requirements using status-update devices and throughput-demand devices. It is shown that when compared with the benchmark scheme [28], in which the maximum allowed AoI constraints are considered, the JOWAC can realize better average AoI performance. The realized AoI performance by the JO scheme also suggests that the maximum allowed AoI constraints by all status-update devices are critical for meeting the strict AoI requirements. Moreover, it is found that the AoI-aware design is more effective than the queue length-aware design in terms of the realized AoI performance.

The remainder of this paper is organized as follows. The system model and problem formulation are presented in Section 1. Adaptive joint-optimization design will be presented in Section 2. Numerical results are presented in Section 3, and we conclude our work in Section 4.

## 2. System Model and Problem Formulation

### 2.1. System Model

As shown in Figure 1, let us consider an uplink multiple-access system with *M* throughput-demand devices and *N* status-update devices. Devices for both types of services transmit their data from upper-layer applications to the access point (AP) in a time-division multiple-access (TDMA) manner. All services are equipped with a data buffer to temporarily store data from either the installed sensors or upper-layer applications. Let $Q_{m}\left(t\right),m\in \left\{1,...,M\right\}$ and $Q_{n}\left(t\right),n\in \left\{1,...,N\right\}$ denote the data buffer states in time slot *t* for throughput-demand device *m* and status-update device *n*, respectively, both of which assume the *first-come, first-served* (FCFS) queuing policy. In addition, we assume that there is a data controller for each throughput-demand device to control its traffic arrival. We assume that the system can acquire perfect channel state information (CSI) for all devices to the AP and that all channels are block fading. Let $h_{m}\left(t\right)$ and $h_{n}\left(t\right)$ denote the channel coefficients from the throughput-demand service device *m* and the status-update service device *n* to the AP in time slot *t*, respectively. We use the binary variable $d_{k}\left(t\right)\in \left\{0,1\right\},k\in \left\{\left\{1,...,M\right\}\cup \left\{1,...,N\right\}\right\}$ to indicate whether device *k* is scheduled to transmit data at time slot *t*. More specifically, $d_{m}\left(t\right)=1$ represents the throughput-demand device, and $d_{n}\left(t\right)=1$ represents the throughput-demand device status-update device at time slot *t*. Due to the use of TDMA transmission mechanism, $\sum_{n=1}^{N}d_{k}\left(t\right)=1,\forall k,t$. The signal received by the AP from device *k* can be expressed as:(1)$$\begin{array}{c} y_{k,AP}\left(t\right)=\sqrt{P_{k}\left(t\right)}h_{k}\left(t\right)x_{k}\left(t\right)+z_{k,AP}\left(t\right),\forall k,t \end{array}$$
where $P_{k}\left(t\right)$ and $x_{k}\left(t\right)$ are the transmit power and transmit signal of device *k* and $E[|x_{k}\left(t\right){|}^{2}]=1$. $z_{k,AP}\left(t\right)$ denotes the Additive White Gaussian Noise (AWGN) at AP and $z_{k,AP}\left(t\right)∼CN\left(0,\sigma_{k}\right)$.

### 2.2. Throughput-Demand Devices

For throughput-demand devices, we assume that the data arrival (bits/slot/Hz) (for instance, from upper-layer applications) obeys the Poisson process. Let $a_{m}\left(t\right)$ denote the amount of data stored in the data buffer in time slot *t*. Let $R_{m}\left(t\right)$ (bits/slot/Hz) stand for the transmission rate of device *m* in time slot *t*. Thus:(2)$$\begin{array}{c} R_{m}\left(t\right)=\log_{2}\left(1+\frac{P_{m}\left(t\right){\left|h_{m}\left(t\right)\right|}^{2}}{\sigma_{m}^{2}}\right),\forall m,t \end{array}$$
where $P_{m}\left(t\right)$ denotes the transmit power of device *m* and $\sigma_{m}^{2}$ denotes the AWGN variance of $y_{m,AP}\left(t\right)$. Meanwhile, the evolution of data buffer $Q_{m}\left(t\right)$ of device *m* in time slot *t* can be expressed as:(3)$$\begin{array}{c} Q_{m}\left(t\;+\;1\right)\;=\;{\left(Q_{m}\left(t\right)\;+\;a_{m}\left(t\right)\;-\;d_{m}\left(t\right)R_{m}\left(t\right)\right)}^{+},\forall m,t \end{array}$$
where ${\left(\cdot \right)}^{+}=\max\left\{\cdot ,0\right\}$.

### 2.3. Status-Update Devices

For the status-update devices, we assume that the arrival of status-update packets (for instance, from the installed sensors) obeys the Bernoulli distribution with parameter $\lambda_{n}$ (packets/slot). Let $a_{n}\left(t\right)\in \left\{0,1\right\}$ denote the number of packets generated at device *n* in time slot *t*. Thus, $E\left[a_{n}\left(t\right)\right]=\lambda_{n}$. Meanwhile, let $L_{n}$ (bits) denote the size of each status-update packet for status-update device *n*. Let $B_{n},P_{n}\left(t\right)$ and $\sigma_{n}^{2}$ stand for the channel bandwidth, transmit power of device *n* and AWGN variance of $y_{n,AP}\left(t\right)$, respectively, so that the transmission rate $R_{n}\left(t\right)$ (bits/slot) can be given by:(4)$$\begin{array}{c} R_{n}\left(t\right)=B_{n}\log_{2}\left(1+\frac{P_{n}\left(t\right){\left|h_{n}\left(t\right)\right|}^{2}}{\sigma_{n}^{2}}\right),\forall n,t \end{array}$$

#### Age of Information

We use Age of Information (AoI) to characterize the data freshness of the status-update devices, and in addition, to make more reasonable use of the underlying physical layer channel capacity to improve the AoI performance, we consider the multi-packet transmission policy. Let indicator function $I_{n}\left(t\right)\in \left\{0,1\right\}$ indicate whether the device *n* can successfully transmit $N_{n}\left(t\right)\in I^{+}$ packets to the AP when device *n* is scheduled by the system. $I_{n}\left(t\right)$ can be expressed as:(5)$$\begin{array}{c} I_{n}\left(t\right)=\left\{\begin{array}{ccc} 1,\; & \; & \mathrm{if}\;d_{n}\left(t\right)R_{n}\left(t\right)\geq N_{n}\left(t\right)L_{n}\\ 0,\; & \; & \mathrm{otherwise}. \end{array}\;,\forall n,t\right. \end{array}$$

Let $u_{n}\left(t\right)$ denote the generation time of the latest packet received by the AP from the status-update device *n* in time slot *t*. Thus, the AoI evolution of device *n* can be expressed as:(6)$$A_{n}\left(t+1\right)=\left\{\begin{array}{ccc} t+1-u_{n}\left(t\right), & \; & \mathrm{if}\;d_{n}\left(t\right)=1,I_{n}\left(t\right)=1,\\ A_{n}\left(t\right)+1, & \; & \mathrm{otherwise}. \end{array}\right.$$
At each time slot *t*, the data buffer state $Q_{n}\left(t\right)$ for the status-update device *n* can be expressed as:(7)$$\begin{array}{c} Q_{n}\left(t\;+\;1\right)\;=\;{\left(Q_{n}\left(t\right)\;+\;a_{n}\left(t\right)L_{n}\;-\;d_{n}\left(t\right)I_{n}\left(t\right)R_{n}\left(t\right)\right)}^{+}\;\;,\forall n,t \end{array}$$

### 2.4. Problem Formulation

Our goal is to maximize the average sum rate of the throughput-demand devices and minimize the average AoI of the status-update devices, subject to the data-queue evolution constraints and peak transmit power constraints for both types of devices, and the average AoI constraint for the status-update devices. The optimization problem can be formulated as follows: $$\begin{array}{cc} P1: & \min_{a_{m}\left(t\right),P_{m}\left(t\right),P_{n}\left(t\right),d_{m}\left(t\right),d_{n}\left(t\right),N_{n}\left(t\right)}\lim_{T\to \infty }\frac{1}{T}\sum_{t=0}^{T-1}\left\{\sum_{n=1}^{N}\theta_{n}A_{n}\left(t\right)-\sum_{m=1}^{M}\theta_{m}a_{m}\left(t\right)\right\}\\ C1: & Q_{m}\left(t\;+\;1\right)\;=\;{\left(Q_{m}\left(t\right)\;+\;a_{m}\left(t\right)\;-\;d_{m}\left(t\right)R_{m}\left(t\right)\right)}^{+}\;\;,\forall m,t,\\ C2: & Q_{n}\left(t\;+\;1\right)\;=\;{\left(Q_{n}\left(t\right)\;+\;a_{n}\left(t\right)L_{n}\;-\;d_{n}\left(t\right)I_{n}\left(t\right)R_{n}\left(t\right)\right)}^{+}\;\;,\forall n,t,\\ C3: & 0\leq a_{m}\left(t\right)\leq {\hat{a}}_{m},\forall m,t,\\ C4: & \lim_{T\to \infty }\frac{1}{T}\sum_{t=0}^{T-1}d_{m}\left(t\right)P_{m}\left(t\right)\leq {\bar{P}}_{m},\forall m,t,\\ C5: & \lim_{T\to \infty }\frac{1}{T}\sum_{t=0}^{T-1}d_{n}\left(t\right)P_{n}\left(t\right)\leq {\bar{P}}_{n},\forall n,t,\\ C6: & 0\leq d_{m}\left(t\right)P_{m}\left(t\right)\leq {\hat{P}}_{m},\forall m,t\\ C7: & 0\leq d_{n}\left(t\right)P_{n}\left(t\right)\leq {\hat{P}}_{n},\forall n,t\\ C8: & \lim_{T\to \infty }\frac{1}{T}\sum_{t=0}^{T-1}d_{n}\left(t\right)I_{n}\left(t\right)R_{n}\left(t\right)\geq \lambda_{n}L_{n},\forall n,t\\ C9: & A_{n}\left(t+1\right)\;=\;\left\{\begin{array}{cc} t+1-u_{n}\left(t\right), & \;\;\;\;\mathrm{if}\;d_{n}\left(t\right)\;=\;1,I_{n}\left(t\right)\;=\;1,\\ A_{n}\left(t\right)+1, & \;\;\;\;\mathrm{otherwise}, \end{array}\right.\;\;,\forall n,t,\\ C10 & :I_{n}\left(t\right)\;=\;\left\{\begin{array}{cc} 1,\; & \;\;\;\;\mathrm{if}\;d_{n}\left(t\right)R_{n}\left(t\right)\geq N_{n}\left(t\right)L_{n},\\ 0,\; & \;\;\;\;\mathrm{otherwise}, \end{array}\right.\forall n,t,N_{n}\left(t\right)\in N^{+},\\ C11 & :\lim_{T\to \infty }\frac{1}{T}\sum_{t=0}^{T-1}A_{n}\left(t+1\right)\leq {\bar{A}}_{n},\forall n,t,\\ C12 & :d_{k}\left(t\right)\in \left\{0,1\right\},\forall k\in \left\{\left\{1,...,M\right\}\cup \left\{1,...,N\right\}\right\},t,\\ C13 & :\sum_{k=1}^{K}d_{k}\left(t\right)=1,\forall k\in \left\{\left\{1,...,M\right\}\cup \left\{1,...,N\right\}\right\},t\; \end{array}$$
where $\theta_{m}\in \left[0,1\right]$ and $\theta_{n}\in \left[0,1\right]$ denote the throughput priority level of throughput-demand devices and AoI priority level of status-update devices, where $\theta_{m}+\theta_{n}=1$, ${\left(x\right)}^{+}=max\left\{0,x\right\}$. C1 and C2 are the data-queue evolution processes for throughput-demand devices and status-update devices, respectively. C3 indicates that the actual amount of data by the throughput-demand devices cannot exceed the maximum value of the upper-layer application. C4 and C5 are the average transmit power constraints for the throughput-demand devices and status-update devices, respectively. C6 and C7 denote the peak transmit power constraints for the throughput-demand devices and the status-update devices, respectively. C8 indicates that the average transmission rate of the status-update devices should be greater than the arrival rate to ensure that C2 is stable. C11 denotes the average AoI constraint for the status-update devices. C12 and C13 denote the transmission mode constraints of the TDMA system. **P1** is a non-convex mixed-integer programming problem, and we will show how to solve it in the next section through the Lyapunov optimization framework and propose an adaptive joint optimization with an AoI-constrained scheme.

## 3. Adaptive Joint Optimization with AoI-Constrained Scheme Design

### 3.1. Lyapunov Optimization Framework

In this section, we use the Lyapunov optimization framework to transform the long-term time-averaged constraint into a problem of queue stability. On this basis, we propose adaptive joint optimization with an AoI-constrained scheme. First, let us transform C4, C5 and C11 into the following virtual AoI evolution queue:(8)$$\begin{array}{cc} \; & Z_{m}\left(t+1\right)={\left(Z_{m}\left(t\right)+d_{m}\left(t\right)P_{m}\left(t\right)-{\bar{P}}_{m}\right)}^{+},\forall m,t, \end{array}$$(9)$$\begin{array}{cc} \; & Z_{n}\left(t+1\right)={\left(Z_{n}\left(t\right)+d_{n}\left(t\right)P_{n}\left(t\right)-{\bar{P}}_{n}\right)}^{+},\forall n,t, \end{array}$$(10)$$\begin{array}{cc} \; & {𝔸}_{n}\left(t+1\right)={\left({𝔸}_{n}\left(t\right)+A_{n}\left(t+1\right)-{\bar{A}}_{n}\right)}^{+},\forall n,t. \end{array}$$

**Theorem** **1.** 
*If all queues are rate stable, i.e., $\lim_{T\to \infty }\frac{Q_{m}\left(T\right)}{T}=\lim_{T\to \infty }\frac{Q_{n}\left(T\right)}{T}=\lim_{T\to \infty }\frac{Z_{m}\left(T\right)}{T}=\lim_{T\to \infty }\frac{Z_{n}\left(T\right)}{T}=\lim_{T\to \infty }\frac{{𝔸}_{n}\left(T\right)}{T}=0$, the constraints C1, C2, C4, C5, C8 and C11 can be satisfied, and then we have:*

(11)
$$\begin{array}{c} \lim_{T\to \infty }\frac{1}{T}\sum_{t=0}^{T-1}d_{m}\left(t\right)R_{m}\left(t\right)\geq \lim_{T\to \infty }\frac{1}{T}\sum_{t=0}^{T-1}a_{m}\left(t\right). \end{array}$$



**Proof.** Please see Appendix A. □

Theorem 1 shows us that if the actual $Q_{m}\left(t\right)$ and Qn are rate stable, the data arrivals of both types of service data queues can be successfully transmitted to the AP. If the virtual power queue $Z_{m}\left(t\right)$ and $Z_{n}\left(t\right)$, and the virtual AoI evolution queue ${𝔸}_{n}\left(t\right)$ are rate stable, it means that the average AoI constraint ${\bar{A}}_{n}$ can be satisfied. Based on this, we define the following quadratic Lyapunov function:(12)$$\begin{array}{c} L\left(\Theta \left(t\right)\right)=\frac{1}{2}\sum_{m=1}^{M}\left\{\mu_{Q,m}Q_{m}^{2}\left(t\right)+\mu_{Z,m}Z_{m}^{2}\left(t\right)\right\}+\frac{1}{2}\sum_{n=1}^{N}\left\{\mu_{Q,n}Q_{n}^{2}\left(t\right)+\mu_{Z,n}Z_{n}^{2}\left(t\right)+\mu_{A,n}{𝔸}_{n}^{2}\left(t\right)\right\}, \end{array}$$
where $\Theta \left(t\right)=\left[Q_{m}\left(t\right),Q_{n}\left(t\right),Z_{m}\left(t\right),Z_{n}\left(t\right),{𝔸}_{n}\left(t\right)\right]$ denotes the queue state vector at the beginning of time slot *t*. $\mu_{Q,m},\mu_{Q,n},\mu_{Z,m},\mu_{Z,n},\mu_{A,n}$ denote the corresponding queue weighting coefficients, respectively, which are employed to ensure that the changes in all queues are in the same order of magnitude. Since the joint queue backlog will increase in the undesired direction, we can introduce a Lyapunov drift representing the change in the Lyapunov function between two consecutive time slots:(13)$$\begin{array}{c} \Delta (\Theta (t))=E[L(\Theta (t+1))-L(\Theta (t))|\Theta (t)], \end{array}$$
where E[·] represents the statistical expectation induced by the control decision at the beginning of each time slot *t*. To ensure that all queues are rate stable, we need to minimize (Equation 13). When ensuring that all constraints are satisfied, we also need to simultaneously maximize the weighted sum rate of throughput-demand devices and minimize the average weighted AoI of status-update devices. Therefore, at the beginning of each time slot, based on the joint queue state Θ(t) of the current time slot, we need to minimize the following Lyapunov drift-plus-penalty value function:(14)$$\Delta \left(\Theta \left(t\right)\right)+VE\left[\sum_{n=1}^{N}\theta_{n}A_{n}\left(t+1\right)-\sum_{m=1}^{M}\theta_{m}a_{m}\left(t\right)|\Theta \left(t\right)\right],$$
where *V* is a non-negative weighting factor that adjusts the tradeoff between the joint queue state and the objective function. Solving (Equation 14) directly is very difficult, and we can solve (Equation 14) by minimizing the upper bound of (Equation 14).

**Theorem** **2.** 
*The upper bound of the Lyapunov drift plus penalty can be given by:*

(15)
$$\begin{array}{cc} \; & \Delta \left(\Theta \left(t\right)\right)+VE\left[\sum_{n=1}^{N}\theta_{n}A_{n}\left(t+1\right)-\sum_{m=1}^{M}\theta_{m}a_{m}\left(t\right)|\Theta \left(t\right)\right]\leq \\ \; & C_{0}+\sum_{m=1}^{M}\left\{\mu_{Q,m}Q_{m}\left(t\right)E\left[a_{m}\left(t\right)-d_{m}\left(t\right)R_{m}\left(t\right)|\Theta \left(t\right)\right]+\mu_{Z,m}Z_{m}\left(t\right)E\left[d_{m}\left(t\right)P_{m}\left(t\right)-{\bar{P}}_{m}|\Theta \left(t\right)\right]\right\}\\ \; & \;\;+\sum_{n=1}^{N}\{\mu_{Q,n}Q_{n}\left(t\right)E\left[a_{n}\left(t\right)L_{n}-d_{n}\left(t\right)I_{n}\left(t\right)R_{n}\left(t\right)|\Theta \left(t\right)\right]+\mu_{Z,n}Z_{n}\left(t\right)E\left[d_{n}\left(t\right)P_{n}\left(t\right)-{\bar{P}}_{n}|\Theta \left(t\right)\right]\\ \; & \;\;+\frac{1}{2}\mu_{A,n}E\left[d_{n}\left(t\right)I_{n}\left(t\right)\left({\left(t+1-u_{n}\left(t\right)\right)}^{2}-{\left(A_{n}\left(t\right)+1\right)}^{2}+2{𝔸}_{n}\left(t\right)\left(t-u_{n}\left(t\right)-A_{n}\left(t\right)\right)\right)|\Theta \left(t\right)\right]\}\\ \; & \;\;+VE\left[\sum_{n=1}^{N}\theta_{n}A_{n}\left(t+1\right)-\sum_{m=1}^{M}\theta_{m}a_{m}\left(t\right)|\Theta \left(t\right)\right], \end{array}$$

*where $C_{0}$ is a constant.*

(16)
$$\begin{array}{cc} C_{0}= & \frac{1}{2}\sum_{m=1}^{M}\left[\mu_{Q,m}\left(a_{m}^{2}+{\hat{R}}_{m}^{2}\right)+\mu_{Z,m}\left({\bar{P}}_{m}^{2}+{\hat{P}}_{m}^{2}\right)\right]+\frac{1}{2}\sum_{n=1}^{N}\{\mu_{Q,n}\left(L_{n}^{2}+{\hat{R}}_{n}^{2}\right)+\mu_{Z,n}\left({\bar{P}}_{n}^{2}+{\hat{P}}_{n}^{2}\right)+\mu_{A,n}[{\bar{A}}_{n}^{2}\\ \; & +{\left(A_{n}\left(t\right)+1\right)}^{2}+2{𝔸}_{n}\left(t\right)\left(A_{n}\left(t\right)+1\right)-2{𝔸}_{n}\left(t\right)\bar{A}]\}. \end{array}$$



**Proof.** Please see Appendix B. □

${\hat{R}}_{m}$ and ${\hat{R}}_{n}$ in (16) represent the maximum transmission rates of throughput-demand devices and status-update devices, respectively.

According to Theorem 2, given the current joint queue state Θ(t) and CSI, in each time slot *t*, **P1** can be effectively solved by the flow control mechanism $a_{m}\left(t\right)$ of throughput-demand devices, the number of transmission packets $N_{n}\left(t\right)$ of status-update devices, and the service mode selection. **P1** can be transformed into the following optimization problem: $$\begin{array}{cc} P2: & \min_{a_{m}\left(t\right),P_{m}\left(t\right),P_{n}\left(t\right),d_{m}\left(t\right),d_{n}\left(t\right),N_{n}\left(t\right)}:\sum_{m=1}^{M}\left\{\mu_{Q,m}Q_{m}\left(t\right)\left[a_{m}\left(t\right)-d_{m}\left(t\right)R_{m}\left(t\right)\right]+\mu_{Z,m}Z_{m}\left(t\right)d_{m}\left(t\right)P_{m}\left(t\right)\right\}\\ \; & +\sum_{n=1}^{N}\{-\mu_{Q,n}Q_{n}\left(t\right)d_{n}\left(t\right)I_{n}\left(t\right)R_{n}\left(t\right)+\mu_{Z,n}Z_{n}\left(t\right)d_{n}\left(t\right)P_{n}\left(t\right)+\frac{1}{2}\mu_{A,n}d_{n}\left(t\right)I_{n}\left(t\right)[{\left(t+1-u_{n}\left(t\right)\right)}^{2}\\ \; & -{\left(A_{n}\left(t\right)+1\right)}^{2}+2{𝔸}_{n}\left(t\right)\left(t-u_{n}\left(t\right)-A_{n}\left(t\right)\right)]\}+V\left[\sum_{n=1}^{N}\theta_{n}A_{n}\left(t+1\right)-\sum_{m=1}^{M}\theta_{m}a_{m}\left(t\right)\right]\\ \; & s.t.\;\;\;\;\;C3,\;C6,\;C7,\;C8,\;C9,\;C10,\;C12,\;C13. \end{array}$$

### 3.2. Flow Control Mechanism

In **P2**, $a_{m}\left(t\right)$ is independent of other optimization variables, so the optimal flow control mechanism can be transformed into the following subproblem:$$\begin{array}{cc} P2.1: & \min_{a_{m}\left(t\right)}:\sum_{m=1}^{M}\left(\mu_{Q,m}Q_{m}\left(t\right)-V\theta_{m}\right)a_{m}\left(t\right)\\ \; & s.t.\;0\leq a_{m}\left(t\right)\leq {\hat{a}}_{m}\forall m,t. \end{array}$$
**P2.1** is a linear programming problem whose optimal solution can be obtained on the boundary, and the optimal flow control mechanism can be given by the following equation:(17)$$\begin{array}{cc} \; & a_{m}\left(t\right)\;=\;\left\{\begin{array}{cc} {\hat{a}}_{m},\; & \;\;\;\;\mathrm{if}\;\mu_{Q,m}Q_{m}\left(t\right)\;<\;V\theta_{m}\\ 0,\; & \;\;\;\;\mathrm{otherwise}. \end{array}\right. \end{array}$$
From (Equation 17), when the backlog of $Q_{m}\left(t\right)$ is small, more data should be placed in the data buffer, and increasing the value of the parameter *V* will also put more data in the data buffer.

### 3.3. Throughput-Demand Devices Transmission Mode

At time slot *t*, when $d_{m}\left(t\right)=1$, the system schedules throughput-demand devices for data transmission. At this time, the throughput-demand devices transmit the data in $Q_{m}\left(t\right)$, and the AoI of all status-update devices will increase. Therefore, the optimization problem **P2** can be transformed into the following subproblem: $$\begin{array}{cc} P2.2:\min_{P_{m}\left(t\right),d_{m}\left(t\right)}\sum_{m=1}^{M}[ & -\mu_{Q,m}Q_{m}\left(t\right)d_{m}\left(t\right)\log_{2}\left(1+\frac{P_{m}\left(t\right){\left|h_{m}\left(t\right)\right|}^{2}}{\sigma_{m}^{2}}\right)+\mu_{Z,m}Z_{m}\left(t\right)d_{m}\left(t\right)P_{m}\left(t\right)]\\ \; & s.t.\;\;C6,\sum_{m=1}^{M}d_{m}\left(t\right)\leq 1. \end{array}$$
One can notice that **P2.2** is a convex problem. According to the KKT condition, the optimal transmit power $P_{m}^{*}\left(t\right)$ of the throughput demanding device can be given by the following equation:(18)$$\begin{array}{cc} \; & P_{m}^{*}\left(t\right)\;=\;\left\{\begin{array}{cc} {\hat{P}}_{m},\; & \;\;\;\;\mathrm{if}\;Z_{m}\left(t\right)=0,\\ {\left\{\frac{\mu_{Q,m}Q_{m}\left(t\right)}{\mu_{Z,m}Z_{m}\left(t\right)ln2}-\frac{\sigma_{m}^{2}}{h_{m}{\left(t\right)}^{2}}\right\}}^{f},\; & \;\;\;\;\mathrm{otherwise}, \end{array}\right. \end{array}$$
where ${\left\{\cdot \right\}}^{f}=min\left\{max\left(\cdot ,0\right),{\hat{P}}_{m}\right\}$. From (Equation 18), we can see that the value of $P_{m}^{*}\left(t\right)$ mainly depends on the CSI of the throughput-demand device *m* and the backlog of data queue $Q_{m}\left(t\right)$. When the CSI is better or $Q_{m}\left(t\right)$ is larger, the system allocates larger transmission power to the device *m*. In addition, $P_{m}^{*}\left(t\right)$ will also be limited by the average transmission power ${\bar{P}}_{m}$, a larger ${\bar{P}}_{m}$ will make $Z_{m}\left(t\right)$ smaller, thus making $P_{m}^{*}\left(t\right)$ larger.

### 3.4. Status-Update Devices Mode

When $d_{n}\left(t\right)=1$, the system schedules the status-update service to transmit data, in which the optimization problem **P2** can be transformed into the following subproblem: $$\begin{array}{cc} P2.3:\min_{P_{n}\left(t\right),d_{n}\left(t\right),N_{n}\left(t\right)}\sum_{n=1}^{N}\{ & -\mu_{Q,n}Q_{n}\left(t\right)d_{n}\left(t\right)I_{n}\left(t\right)R_{n}\left(t\right)+\mu_{Z,n}Z_{n}\left(t\right)d_{n}\left(t\right)P_{n}\left(t\right)+\frac{1}{2}\mu_{A,n}d_{n}\left(t\right)I_{n}\left(t\right)\times \\ \; & \left[{\left(t+1-u_{n}\left(t\right)\right)}^{2}-{\left(A_{n}\left(t\right)+1\right)}^{2}+2{𝔸}_{n}\left(t\right)\left(t-u_{n}\left(t\right)-A_{n}\left(t\right)\right)\right]\}\\ \; & +V\theta_{n}\left(t\;+\;1\;-\;u_{n}\left(t\right)\right)\;+\;V\sum_{n^{\prime }\neq n}\theta_{n^{\prime }}\left(A_{n^{\prime }}\left(t\right)\;+\;1\right)\\ \; & s.t.\;\;\left(4\right),C7,\;C10,\;\sum_{n=1}^{N}d_{n}\left(t\right)\leq 1. \end{array}$$

Therefore, the optimal transmission power $P_{n}^{*}\left(t\right)$ of status-update device *n* at time slot *t* can be given by:(19)$$\begin{array}{cc} \; & P_{n}^{*}\left(t\right)\;=\;\left\{\begin{array}{cc} \min\{\frac{\sigma_{n}^{2}\left(2^{\frac{N_{n}\left(t\right)L_{n}}{B_{n}}}-1\right)}{h_{n}{\left(t\right)}^{2}},\hat{P}\},\; & \;\;\;\;\mathrm{if}\;P_{n}\left(t\right)\leq {\hat{P}}_{n},\\ 0,\; & \;\;\;\;\mathrm{otherwise}. \end{array}\right. \end{array}$$
As can be seen in (Equation 19), unlike the power optimization for throughput-demand devices, the worse the channel state of the status-update device *n* to the AP communication link, the larger the transmit power $P_{n}^{*}\left(t\right)$ allocated by the system to enable packets being transmitted as successfully as possible. In addition, $P_{n}^{*}\left(t\right)$ also depends on the numbers of packets $N_{n}\left(t\right)$ that the status-update device *n* can transmit at the current time slot *t*. The optimal number of packets $N_{n}^{*}\left(t\right)$ to be transmitted by device *n* at time slot *t* can be given by:(20)$$\begin{array}{cc} N_{n}^{*}\left(t\right)=\arg\min_{N_{n}=1,2,...,\hat{N_{n}}} & -\mu_{Q,n}Q_{n}\left(t\right)N_{n}\left(t\right)L_{n}+\mu_{Z,n}Z_{n}\left(t\right)P_{n}^{*}\left(t\right)+\frac{1}{2}\mu_{A,n}[{\left(t+1-u_{n}\left(t\right)\right)}^{2}-{\left(A_{n}\left(t\right)+1\right)}^{2}\\ \; & +2{𝔸}_{n}\left(t\right)\left(t-u_{n}\left(t\right)-A_{n}\left(t\right)\right)]+V\theta_{n}\left(t\;+\;1\;-\;u_{n}\left(t\right)\right)\;+\;V\sum_{n^{\prime }\neq n}\theta_{n^{\prime }}\left(A_{n^{\prime }}\left(t\right)\;+\;1\right), \end{array}$$
where $\hat{N_{n}}$ is the maximum number of packets that status-update device *n* can transmit at time slot *t*. $\hat{N_{n}}$ is limited by the current backlog of data queue $Q_{n}\left(t\right)$ of device *n* and the peak transmit power ${\hat{P}}_{n}$. $\hat{N_{n}}$ can be expressed as:(21)$$\begin{array}{c} \hat{N_{n}}=\min\left\{\;\frac{Q_{n}\left(t\right)}{L_{n}},\left\lfloor \;\frac{B_{n}\log_{2}\left(1\;+\;\frac{{\hat{P}}_{n}h_{n}{\left(t\right)}^{2}}{\sigma_{n}^{2}}\right)}{L_{n}}\;\right\rfloor \;\right\}. \end{array}$$
Substituting (Equation 19) into (Equation 20) yields the number of packets transmitted by device *n* at time slot *t*, as well as the optimal transmit power $P_{n}^{*}\left(t\right)$. In (Equation 20), it is shown that the status-update device always prioritizes the transmission of $\hat{N_{n}}$ packets, but due to the effect of the average power constraint (i.e., the term of $\mu_{Z,n}Z_{n}\left(t\right)P_{n}^{*}\left(t\right)$), status-update device does not always transmit $\hat{N_{n}}$ packets at each time slot *t*.

### 3.5. Heterogeneous Service Transmission Mode Selection

At time slot *t*, when the optimal flow control mechanism for the throughput-demand devices, the optimal number of transmitted packets for the status-update devices, and the optimal transmit power allocation for two types of devices are obtained, the transmission mode selection can be determined as follows:(22)$$\begin{array}{c} d_{k}\left(t\right)=\left\{\begin{array}{cc} 1, & \mathrm{if}\;k=\arg\min_{k\in \left\{\left\{1,...,M\right\}\cup \left\{1,...,N\right\}\right\}}\left[L_{m}^{*}\left(t\right),L_{n}^{*}\left(t\right)\right]\\ 0, & \mathrm{otherwise}, \end{array}\right. \end{array}$$
where $L_{m}^{*}\left(t\right)$ and $L_{n}^{*}\left(t\right)$ correspond to the optimal values obtained in **P2.2** and **P2.3** as follows: $$\begin{array}{cc} \; & L_{m}^{*}\left(t\right)=-\mu_{Q,m}Q_{m}\left(t\right)\log_{2}\left(1+\frac{P_{m}^{*}\left(t\right)h_{m}{\left(t\right)}^{2}}{\sigma_{m}^{2}}\right)+\mu_{Z,m}Z_{m}\left(t\right)P_{m}^{*}\left(t\right)+V\sum_{n=1}^{N}\theta_{n}\left(A_{n}\left(t\right)+1\right), \end{array}$$(23)$$\;\;\;\;\;\;\;\;\;\;\;\;\;\;\;\;\forall m\in \left\{1,\ldots ,M\right\},$$
$$L_{n}^{*}\left(t\right)=-\mu_{Q,n}Q_{n}\left(t\right)N_{n}^{*}\left(t\right)L_{n}+\mu_{Z,n}Z_{n}\left(t\right)P_{n}^{*}\left(t\right)+\frac{1}{2}\mu_{A,n}[{\left(t+1-u_{n}\left(t\right)\right)}^{2}-{\left(A_{n}\left(t\right)+1\right)}^{2}$$(24)$$\begin{array}{cc} \; & \;\;+2{𝔸}_{n}\left(t\right)\left(t-u_{n}\left(t\right)-A_{n}\left(t\right)\right)]+V\theta_{n}\left(t\;+\;1\;-\;u_{n}\left(t\right)\right)\;+\;V\sum_{n^{\prime }\neq n}\theta_{n^{\prime }}\left(A_{n^{\prime }}\left(t\right)\;+\;1\right)\;\forall n\in \left\{1,\ldots N\right\}. \end{array}$$
According to (Equation 23) and (24), one can observe that the backlog of data queues, the average transmission power constraint, and the AoI of the status-update service collectively affect the selection of transmission mode for heterogeneous services. For throughput-demand devices, the system will always select the device with a larger backlog of data queue or larger transmission rate, but selecting a throughput-demand device will result in an AoI increase for all status-update devices. Like throughput-demand devices, status-update devices also prioritize devices with larger data-queue backlogs or those that can transmit more packets, and additionally, devices are constrained by the AoI evolution queue ${𝔸}_{n}\left(t\right)$ at the current time slot *t*, which becomes smaller when the AoI demand by device *n* is stringent.

## 4. Numerical Results

In this section, we will illustrate the averaged AoI and average sum-rate performance. In all numerical analysis, the following simulation parameters are assumed (Table 1):

In this paper, the channel coefficient from device *k* to AP is assumed to be given by
(25)$$\begin{array}{c} h_{k}\left(t\right)=\sqrt{10^{-3}D_{k}^{-\tau }}\left[\alpha_{1}\left(t\right),\alpha_{2}\left(t\right),\cdots ,\alpha_{k}\left(t\right)\right] \end{array}$$
where $\alpha_{k}\left(t\right)$ obeys the Rayleigh distribution. All simulation results duration are $10^{6}$ time slots. To better explore the performance of the proposed JOWAC scheme, we include the joint optimization without the AoI constraint scheme (JO) and the weighted sum-rate optimization scheme (WSRO) [28] in the following numerical analysis. To better illustrate the superiority of the transmission schemes that we proposed in this paper, we further consider two benchmark schemes: data-queue length priority scheme (QLP) and queue length Round-Robin scheme (QLRR). In the QLP scheme, the scheduling for device transmission will consider the data-queue lengths (unit: bits) of the two types of devices, i.e., the device with the largest data-queue backlog in the current time slot *t* will always have the transmission opportunity. Instead, QLRR is a polling-based scheduling scheme that is independent of the data-queue length, i.e., starting from the first throughput-demand device to the last throughput-demand device, then starting from the first status-update device to the last status-update device as a round, scheduling one device for each time slot, and then re-scheduling the first throughput-demand device, and so on until the termination of the last time slot.

Figure 2 illustrates the realized-average AoI performance of all status-update devices and the realized-average sum-rate performance of all throughput-demand devices with different status-update arrival rate λ (packets/slot) for the JOWAC scheme, the WSRO scheme, the JO scheme, the QLP scheme, and the QLRR scheme. The average AoI performance results show a trend of first increasing, then decreasing and later increasing with the increase of arrival rate λ. This is because when λ is low, the larger length packet generation interval of the status-update devices leads to their undesirable average AoI performance, and at this time, decreasing $\bar{A}$ can hardly make the average AoI smaller. As the arrival rate λ increases, the average AoI performance starts to improve and gradually reaches the optimum; when λ further increases, the high backlog of data queue $Q_{n}\left(t\right)$ using the status-update devices makes the average AoI performance deteriorate. At this time, reducing $\bar{A}$, the system will make status-update devices satisfy their average AoI constraint at the cost of reducing the average sum rate. Second, increasing λ makes the system tend to allocate more transmission time slots to the status-update devices, and thus, the average sum rate decays rapidly with the increase in λ. It is worth noting that at high arrival rates (starting from λ>0.45), the JOWAC scheme not only outperforms the WSRO scheme in terms of the average AoI but also slightly outperforms the WSRO scheme in terms of average sum-rate performance for the same $\bar{A}$. This indicates that the JOWAC scheme has an overall performance advantage within the high arrival-rate region. Finally, the adaptive transmission scheme proposed in this paper can make up more reasonable scheduling decisions by rationally utilizing the CSI, PSI, and ASI, as well as the data-queue backlog state information (BSI) of the two types of services in each time slot *t*. Therefore, compared with the QLP scheme and the QLRR scheme, the JOWAC scheme, the WSRO scheme, and the JO scheme can provide larger arrival rate λ support for status-update devices. Moreover, the average AoI performance of the JOWAC scheme, WSRO scheme, and JO scheme always outperforms those of the QLP scheme and QLRR scheme, while the average sum-rate performance of the JOWAC scheme, WSRO scheme, and JO scheme if λ>0.2. This is because when λ<0.2, the $Q_{n}\left(t\right)$ backlog is very small, and the QLP scheme allocates almost all the transmission time slots to the throughput-demand devices. In brief, the JOWAC scheme can realize better average AoI and average sum-rate performance since it can reasonably adjust the transmission scheduling according to BSI $Q_{n}\left(t\right)$. In addition, since the polling scheduling mechanism by the QLRR scheme is not affected by the underlying CSI, PSI, AI, and BSI, its average sum-rate performance remains almost unchanged, but this comes at the cost of the deteriorated average AoI performance.

In Figure 3, we show the realized-average AoI and average sum-rate performance with a different number of throughput-demand devices *M* and different status-update arrival rate λ using JOWAC, WSRO, and JO schemes. As shown in Figure 3a, with a high arrival rate (λ=0.65), the average sum-rate performance and the average AoI performance of the remaining schemes, except for the JO scheme, remain almost unchanged as *M* increases. This is because the JO scheme does not consider the average AoI constraint, and the total number of transmission slots obtained by the throughput-demand devices increases as *M* increases. Hence, in this case, the average AoI will degrade, and the average sum rate will also increase slightly as *M* increases. On the other hand, both the JOWAC scheme and the WSRO scheme consider the average AoI constraint ${\bar{A}}_{n}$. One can observe from Figure 3a that the realized-average AoI performance of both JOWAC and JO schemes stays below ${\bar{A}}_{n}$ and remains unchanged as *M* increases. The realized-average sum-rate performance is almost unchanged as well, which can be explicated by the fact that even though *M* increases, to make sure that the average AoI constraints can be satisfied, the system must allocate (reserve) enough transmission opportunities for status-update devices. As a result, the average sum-rate performance will remain unchanged because the throughput-demand devices cannot have more transmission opportunities with the increase of *M*. However, the increase of *M* will decrease the average rate allocated to every single throughput-demand device. Within a low to moderate arrival-rate region (λ=0.25,λ=0.1), as shown in Figure 3b,c, the average AoI degrades, and the average sum rate improves with the increase in *M*. This is because, within a low to moderate arrival-rate region, the backlog of the data queue $Q_{n}\left(t\right)$ of the status-update devices is not severe, and the average AoI can be ensured to be below ${\bar{A}}_{n}$. Therefore, the system will try to allocate more transmission resources to throughput-demand devices. Basically, due to the increase in *M*, the transmission opportunity obtained by every single throughput-demand device decreases, and the backlog size $Q_{m}\left(t\right)$ increases, which leads to a decrease in the amount of extra data put into $Q_{m}\left(t\right)$ by the upper-layer application. In fact, the amount of backlog size $Q_{m}\left(t\right)$ will gradually approach the stabilization value when *M* is large enough. Compared with the JOWAC scheme, because there is no average AoI constraint in the JO scheme, its realized-average AoI and average sum-rate performance will exhibit a more noticeable increase.

Similarly, we illustrate the realized-average AoI and average sum-rate performance with different numbers of status-update devices *N* and different status-update arrival rates λ in Figure 4. As shown in Figure 4a, within high arrival rates (λ=0.5), a slight increase in *N* makes the average AoI tend to infinity and the average sum rate drop to almost zero. This happens when the system has already consumed almost all available resources to support the critical transmission requirements by the existing status-update devices. In extreme cases, the system cannot afford one more status-update device with a strict average AoI requirement, which leads to scheduling failure. Basically, the JO scheme can support more status-update devices, but it does not ensure the average AoI requirement for every status-update device. As shown in Figure 4b,c, within low to moderate arrival-rate region (λ=0.25,λ=0.1), both the average AoI and the average sum-rate performance degrades as *N* increases. The WSRO scheme can satisfy the average AoI constraint ${\bar{A}}_{n}$, even if it does not consider the average AoI in the optimization metric. This is because the JO scheme does not consider the AoI constraint. It tends to allocate more transmission opportunities for the throughput-demand device, which makes it capable of realizing reasonable average sum-rate performance. This phenomenon will become more obvious at low arrival rates (λ=0.1). The JOWAC scheme considers both the average AoI constraint and the AoI optimization objective. As expected, it can always realize the optimal average AoI performance in all cases. As *N* increases, the realized AoI performance degrades as well, since now more update status devices need to share the transmission opportunities to fulfill their critical freshness requirements. As a result, there will be a worse average sum-rate performance for the throughput-demand devices. Finally, we can conclude from Figure 3 and Figure 4 that the JOWAC can always realize the best average AoI performance, regardless of the choice of *M* and *N* in all status-update arrival-rate regions, which confirms that the introduction of average AoI optimization metric and considering the average AoI constraint are necessary when there are critical AoI performance requirements. Because strict resource demands are required to ensure AoI performance, there will be some loss in the average sum-rate performance for the throughput-demand devices, just as we can expect for a heterogeneous access network.

To clearly highlight the tradeoff between the average AoI performance of the status-update devices and the average sum-rate performance of the throughput-demand devices, we illustrate the AoI-sum-rate tradeoff of the JOWAC scheme, the WSRO scheme, and the JO scheme in Figure 5. For the JOWAC scheme and the JO scheme, $\theta_{m}$ and $\theta_{n}$ vary in the opposite direction, namely the direction of increase along the x-axis and y-axis is the direction in which $\theta_{m}$ increases and $\theta_{n}$ decreases, respectively, i.e., the leftmost corresponds to $\left[\theta_{m}=0.1,\theta_{n}=0.9\right]$, while the rightmost is $\left[\theta_{m}=0.9,\theta_{n}=0.1\right]$. For the WSRO scheme, there is only one weighting coefficient of $\theta_{m}$, the leftmost is $\theta_{m}=0.1$, while the rightmost is $\theta_{m}=0.9$. In heterogeneous status-update and throughput-demand services access networks, we always expect to realize as small as possible AoI and as much as possible sum rate. The leftmost and steepest tradeoff curve is desired. One may readily observe that, except for the large status-update rate case (i.e., λ=0.5), the proposed JOWAC scheme can achieve the best tradeoff. Since the WSRO scheme does not consider the AoI optimization metric, its tradeoff is noticeably inferior to the JOWAC scheme. Compared to the JO scheme, since both the JOWAC scheme and the WSRO scheme consider the AoI constraint, they will try their best to ensure the realized AoI is below the AoI constraint limit at the cost of the reduction in the realized sum-rate performance, as illustrated in Figure 5a. Although the JO scheme seems to be able to realize reasonable tradeoffs in large status-update arrival rate (i.e., λ=0.65), it cannot guarantee the realized AoI performance.

## 5. Conclusions

In this paper, we study a buffer-aided uplink TDMA network, which consists of status-update devices and throughput-demand devices. A joint-optimization problem was formulated to maximize the average sum rate of the throughput-demand devices and to minimize the average AoI of the status-update devices while satisfying the data-queue state evolution constraint and peak transmit power constraint for both types of devices, as well as the average AoI constraints for status-update devices. By solving the problem with the Lyapunov optimization framework, we obtain the AoI-aware adaptive TDMA uplink system. Our analysis results show that the AoI-aware adaptive TDMA uplink scheme can effectively fulfill the heterogeneous service requirements by status-update devices and throughput-demand devices. Meanwhile, the joint optimization of the average sum rate and the average AoI can realize a better sum rate and average AoI performance. The analysis in this paper sheds some light on the buffer-aided multiple-access network with heterogeneous traffic requirements. The accommodation of more devices and consideration of the impact of imperfect channel state information for practical networks will be left for future exploration.

## Figures and Tables

**Figure 1 sensors-24-00506-f001:**
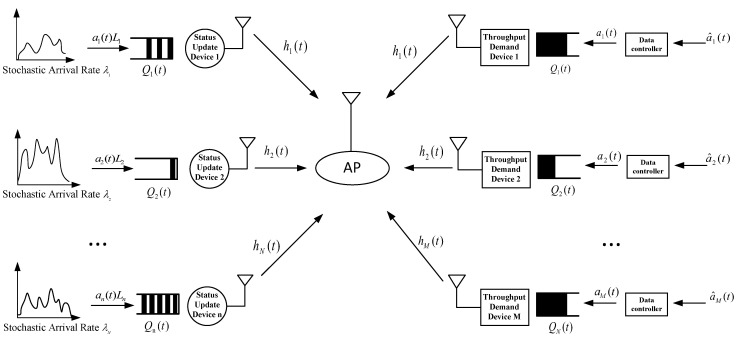
System Model.

**Figure 2 sensors-24-00506-f002:**
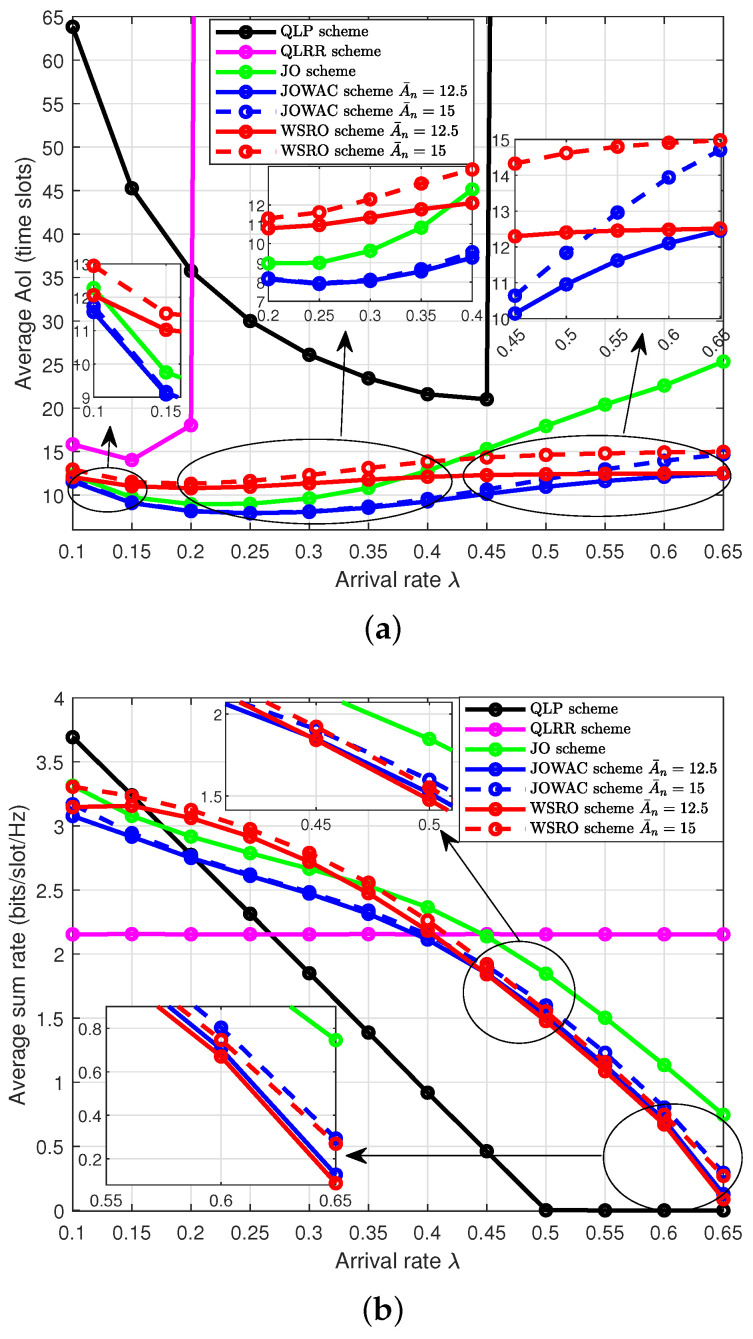
Average AoI and average sum-rate performance with different status-update arrival rates λ, M=N=4, $\theta_{m}=\theta_{n}=0.5$. (**a**) Average AoI performance with different status-update arrival rates; (**b**) Average sum-rate performance with different status-update arrival rates.

**Figure 3 sensors-24-00506-f003:**
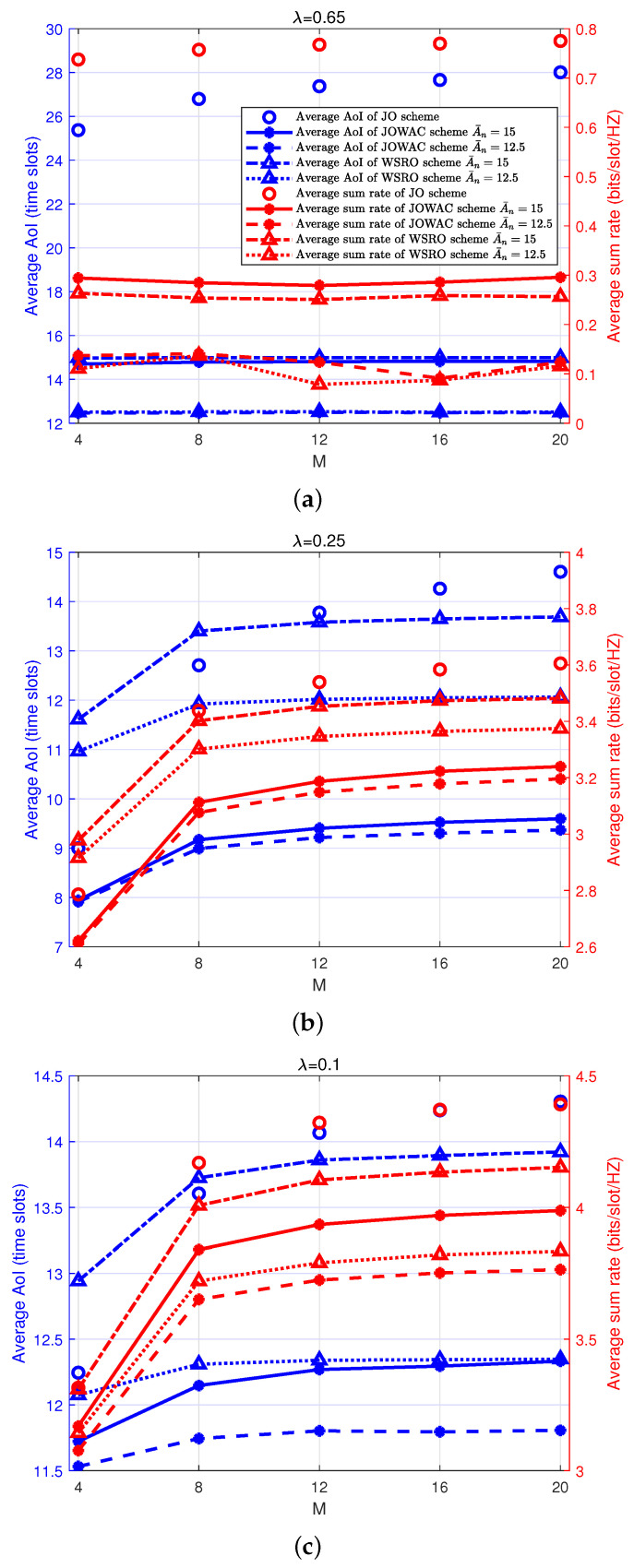
Realized-average AoI and average sum-rate performance with different number of throughput-demand devices and different status-update arrival rate λ, N=4, $\theta_{m}=\theta_{n}=0.5$, where the same legends are assumed for subfigure (**a**–**c**).

**Figure 4 sensors-24-00506-f004:**
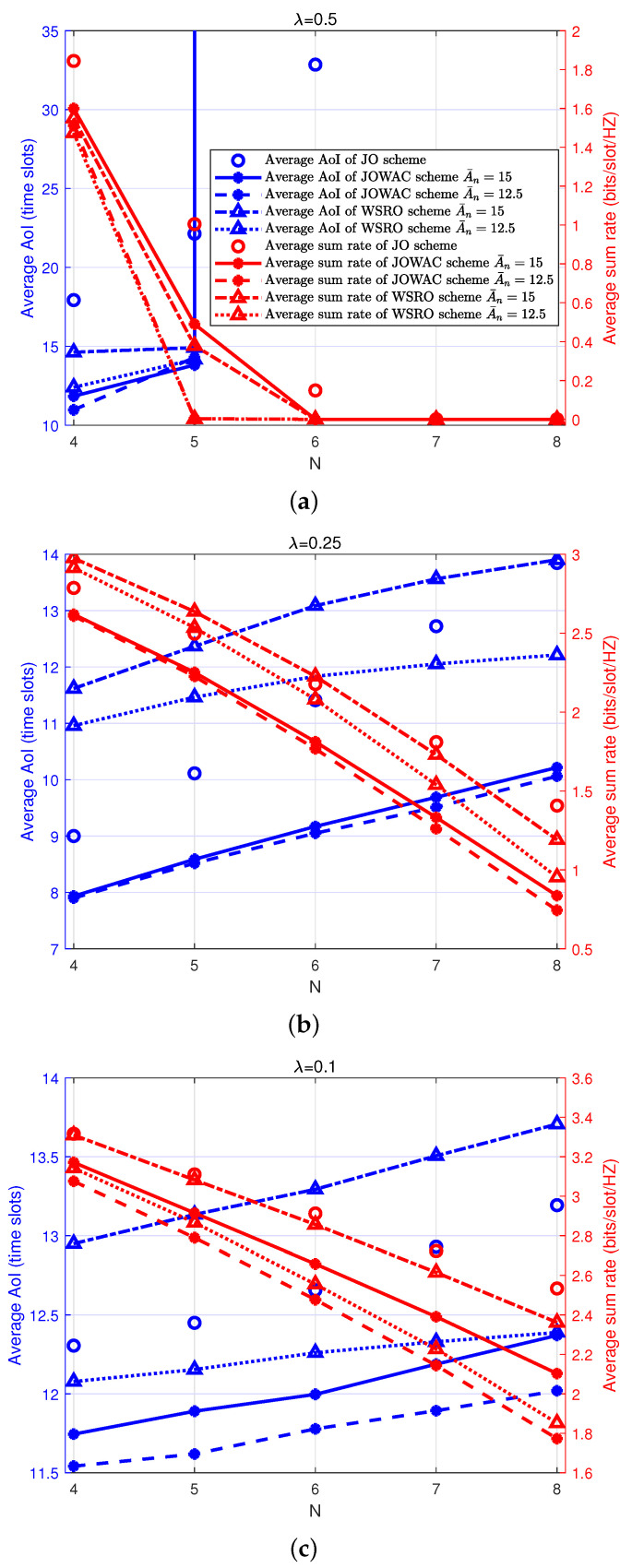
Realized-average AoI and average sum-rate performance with a different number of status-update devices *N* and different status-update arrival rate λ, M=4, $\theta_{m}=\theta_{n}=0.5$, where the same legends are assumed for subfigure (**a**–**c**).

**Figure 5 sensors-24-00506-f005:**
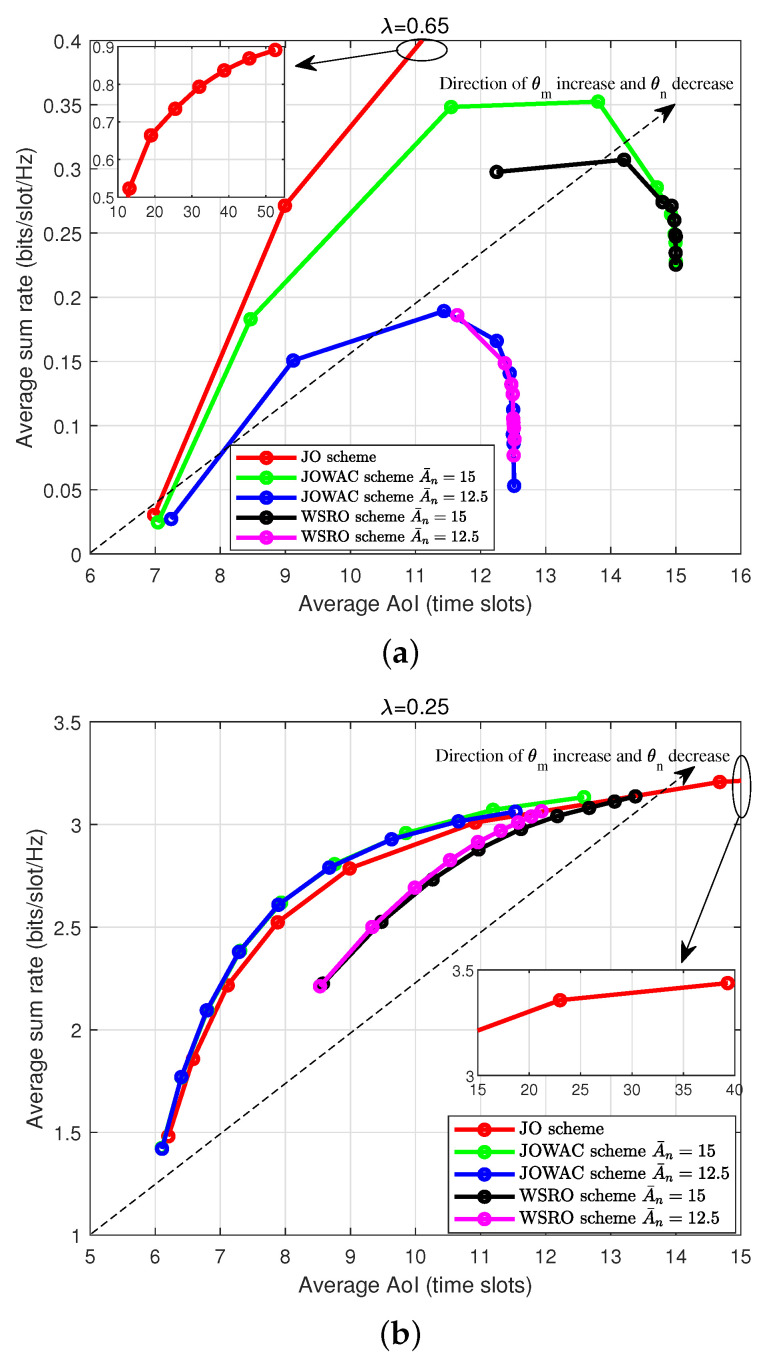
Tradeoffs between average AoI and average sum rate as a function of the weights of the two types of operations, M=N=4.

**Table 1 sensors-24-00506-t001:** Parameters Notation Table.

Parameters	Statement	Statement
$\mu_{Z,m},\mu_{Z,n},\mu_{A,n}$	The weighting coefficients of virtual power queues and AoI virtual queues	$10^{3}$
$\mu_{Q,m},\mu_{Q,n}$	The weighting coefficients of data queues	$10^{4}$
${\bar{P}}_{m},{\bar{P}}_{n}$	The average transmit power	30 dBm
${\hat{P}}_{m},{\bar{P}}_{n}$	The peak transmit power, where ${\hat{P}}_{m}=10\;{\bar{P}}_{m}$ and ${\hat{P}}_{n}=5\;{\bar{P}}_{n}$	300 dBm, 150 dBm
$\sigma_{m},\sigma_{n}$	Noise variances	−20 dBm
$B_{n}$	The transmission bandwidth of status-update devices	1000 Hz
$L_{n}$	The packet sizes of status-update devices	1500 bits
$D_{k}^{-\tau }$	Distance from device $k,k(\;k\in \left\{\left\{1,...,M\right\}\cup \left\{1,...,N\right\}\right\}\;)$ to AP	5 m
τ	path loss factor	2

## Data Availability

Data are contained within the article.

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
