# Peer review of "On the Adaptive Buffer-Aided TDMA Uplink System with AoI-Aware Status-Update Services and Timely Throughput Trafficsâ€"

_sensors, 2024, doi:10.3390/s24020506_

Round 1
Reviewer 1 Report
Comments and Suggestions for Authors
The paper is well organized and easy to follow. I just have some minor comments.
- Try to delete the blank in page 5.
- In the parameter table, there should be a space between "30" and "dBm", so does the rest.
- There are too many subfigures in some figures. It is suggested to draw a bigger figure to show more detail.
Comments on the Quality of English LanguageI have no comments.
Author Response
Reply Letter to the Reviewers
Reviewer 1
Comments and Suggestions for Authors
The paper is well organized and easy to follow. I just have some minor comments.
Comment 1-1:Try to delete the blank in page 5.
Reply to Comment 1-1:
The authors would like to thank the reviewer for the careful reading. We have removed the mentioned blank space on page five.
Comment 1-2:ln the parameter table, there should be a space between "30" and "dBm", so does the rest.
Reply to Comment 1-2:
The authors would like to thank the reviewer for this suggestion. We have added spaces for numbers and units in the parameter table
Comment 1-3:There are too many subfigures in some figures. It is suggested to draw a bigger figure to show more detail
Reply to Comment 1-3:
The authors would like to thank the reviewer for this advice. We have integrated the average AoI of status update devices and the average sum rate of throughput demand devices for the three status update arrival rates in Figures 3 and 4 into one image, respectively. Additionally, we have redrawn Figures 5 to make it more show more details.

Reviewer 2 Report
Comments and Suggestions for Authors
The authors utilize the average age of information to characterize the data freshness of the status update devices, and the average sum-rate to capture the average transmission performance of the throughput demand devices. The formulate a joint optimization problem to minimize the average AoI for status update devices and to maximize the average sum-rate for the throughput demand devices.
What is the simulation tool that you used?
What is the confidence interval?
The authors refer to many papers that they have published. Please explain clearly the contribution of this manuscript compared to your previous published papers.
It is difficult to read the text in the figures, for example, Figure 1.
Comments on the Quality of English LanguageThe authors have used very long sentences. Even some of the sentences do not connect well or make sense. Review is needed.
Example:
This paper considers a buffer-aided TDMA uplink system composed of status-update devices with average AoI requirements and throughput-demand devices with average sum-rate performance requirements. And a joint optimization problem was formulated to maximize the average sum-rate of the throughput demand devices and to minimize the average AoI of the status update devices, while satisfying data queue state evolution constraint and peak transmit power constraint for both types of devices.
Author Response
Reviewer 2
Comments and Suggestions for Authors
The authors utilize the average age of information to characterize the data freshness of the status update devices, and the average sum-rate to capture the average transmission performance of the throughput demand devices. The formulate a joint optimization problem to minimize the average Aol for status update devices and to maximize the average sum-rate for the throughput demand devices.
Comment 2-1:What is the simulation tool that you used?
Reply to Comment 2-1:
In our numerical analysis, we use Matlab as simulation tool.
Comment 2-2:What is the confidence interval?
Reply to Comment 2-2:
The authors would like to thank the reviewer. But there is use of confidence interval in our manuscript.
In our numerical analysis, all results are obtained through 106 time slots.
Comment 2-3:The authors refer to many papers that they have published. Please explain clearly the contribution of this manuscript compared to your previous published papers.
Reply to Comment 2-3:
The authors would like to appreciate the comments by the reviewer.
In previous work, many research results have demonstrated that buffer-aided transmission techniques can effectively improve transmission throughput in various communication system setups. References [21] and [22] studied a buffer-aided wireless energy bidirectional relay communication system, in which [22] highlighted the influence of eavesdroppers on the basis of [21], and proved that the average achievable rate and average achievable secrecy rate of the wireless energy bidirectional relay communication system can be significantly improved when fully considering the potential of data buffer and energy storage. Reference [23] studied a buffer-aided wireless powered uplink multiple access system. In order to reveal the average achievable rate region of limited energy storage and data storage uplink multiple access wireless powered communication systems, the design of downlink energy beamforming, device flow control mechanism, transmission mode selection, device transmit power, and time allocation were jointly optimized to maximize the average weighted sum rate of devices. The analysis results showed that, compared with traditional half-duplex wireless powered communication network and full-duplex wireless powered communication network without buffer-aided mechanism, the buffer-aided adaptive transmission scheme can effectively utilize time diversity gain to obtain a higher average achievable rate region. Reference [24] studied a buffer-aided wireless energy harvesting relay network with imperfect channels, where the relay is provisioned with energy storage and data buffer, and proposed a robust relay transmission scheme, which is shown to be superior to traditional schemes. Reference [25] studied a buffer-aided MISO downlink time division multiple access wireless communication system, where AP is assumed to be equipped with multiple antennas on and the device is with a single antenna. Through parallel flow control mechanism, information signal and artificial noise beamforming, and user selection, we can maximize the long-term average achievable secrecy transmission rate region, and realize a better long-term average achievable secrecy transmission rate region compared to traditional no buffer-aided scheme.
In our previous research work, we have comprehensively investigated how buffer-aided technology can effectively improve the transmission rates of various communication systems. However, the coexistence of multiple heterogeneous devices will present in the future Internet of Things. Exploring the potential of buffer-aided transmission technology and applying it to heterogeneous IoT network to improve its transmission performance is an important research direction. In reference [26], we studied a three-node unidirectional relay network equipped with both throughput demand device and status update device, where data buffers are assumed at both the source and relay nodes. In reference [27], a buffer-aided uplink multiple access wireless communication system was studied for both types of services. The analysis results of both recent literature indicate that, given the average and peak transmit power, the arrival rate of status update services and their long-term average AoI constraints play an important role in the average achievable throughput performance for the throughput demand services. Moreover, if a certain average AoI constraint can be tolerated for status update services, throughput demand devices can achieve better average transmission throughput.
The research contribution of this manuscript is to extend the WSRO scheme proposed in reference [27]. In order to reveal the achievable average AoI performance for status update devices, we propose JOWAC schemes with average AoI constraints and JO schemes without average AoI constraints. At the same time, we compare the realized performance with traditional queue length scheduling schemes and polling scheduling schemes, clearly showing the superiority of the proposed JOWAC schemes. In addition, compared to [27], we conducted in-depth analysis on the impact of the number of two types of devise and the inherent tradeoff between the average AoI performance and the average sum rate performance.
The contributions of this manuscript are briefly summarized on page 3 of the revised manuscript.
- Lan, Q. Chen, X. Tang and L. Cai, "Achievable Rate Region of the Buffer-Aided Two-Way Energy Harvesting Relay Network," in IEEE Transactions on Vehicular Technology, vol. 67, no. 11, pp. 11127-11142, Nov. 2018.
- Nie et al., "Achievable Rate Region of Energy-Harvesting Based Secure Two-Way Buffer-Aided Relay Networks," in IEEE Transactions on Information Forensics and Security, vol. 16, pp. 1610-1625, 2021.
- Lan, Q. Chen, L. Cai and L. Fan, "Buffer-Aided Adaptive Wireless Powered Communication Network With Finite Energy Storage and Data Buffer," in IEEE Transactions on Wireless Communications, vol. 18, no. 12, pp. 5764-5779, Dec. 2019.
- Ni, T. Wang, S. Wang and Q. Chen, "On the Wireless Powered Buffer-Aided Relay Communication with Imperfect CSI," 2023 8th International Conference on Computer and Communication Systems (ICCCS), Guangzhou, China, 2023.
- Lan, J. Ren, Q. Chen and L. Cai, "Achievable Secrecy Rate Region for Buffer-Aided Multiuser MISO Systems," in IEEE Transactions on Information Forensics and Security, vol. 15, pp. 3311-3324, 2020.
- Wang, T. Wang, L. Zheng and Q. Chen, "Adaptive Buffer-aided Wireless Relay Communications with Mixed Status Update and Throughput Traffic," 2023 8th International Conference on Computer and Communication Systems (ICCCS), Guangzhou, China, 2023.
- Wang, S. Wang, L. Zheng and Q. Chen, "Adaptive Transmission Design in TDMA Uplink System with Status Update and Throughput Services," 2023 8th International Conference on Computer and Communication Systems (ICCCS), Guangzhou, China, 2023.
Comment 2-4:It is difficult to read the text in the figures, for example, Figure 1.
Reply to Comment 2-4:
The authors would like to thank the reviewer for this advice. We have revised all figures in the revised manuscript so that they can be recognized more clearly.
Comments on the Quality of English Language
The authors have used very long sentences. Even some of the sentences do not connect well or make sense. Review is needed.
Example:
This paper considers a buffer-aided TDMA uplink system composed of status-update devices with average Aol requirements and throughput-demand devices with average sum-rate performance requirements. And a joint optimization problem was formulated to maximize the average sum-rate of the throughput demand devices and to minimize the average Aol of the status update devices. while satisfying data queue state evolution constraint and peak transmit power constraint for both types of devices.
Reply to Comments on Quality of English language:
The authors would like to thank the reviewer for this advice. We have revised manuscript by following the reviewer’s comment. More specifically, we have tried to avoid using very long sentence, instead, we rephrased with short ones to make the readers easier to follow.

Reviewer 3 Report
Comments and Suggestions for Authors
Dear Editor,
This paper studied a buffer-aided TDMA uplink network, where multiple status update devices and throughput demand devices are supposed to upload their data to one information access point (AP), and all devices are assumed to be provisioned with data buffer to temporarily store the
randomly generated data from either the installed sensor or the upper-layer applications. This paper is interesting, however there are some problems that existed in this paper.
Detailed Issues:
1. This paper investigated the TDMA topic, however, it is very old topic, why the authors investigated this topic, the detailed reasons should be added into the revised manuscript.
2.When using the buffer technology, how to solve the delay problem when using this technology.
3. The system model figure and simulation figures needed to be re-plotted. Especially, where are the simulation results in the simulation figures?
4.Some mistakes and error problems should be corrected in the revised manuscript.
Comments on the Quality of English LanguageDear Editor,
This paper studied a buffer-aided TDMA uplink network, where multiple status update devices and throughput demand devices are supposed to upload their data to one information access point (AP), and all devices are assumed to be provisioned with data buffer to temporarily store the
randomly generated data from either the installed sensor or the upper-layer applications. This paper is interesting, however there are some problems that existed in this paper.
Detailed Issues:
1. This paper investigated the TDMA topic, however, it is very old topic, why the authors investigated this topic, the detailed reasons should be added into the revised manuscript.
2.When using the buffer technology, how to solve the delay problem when using this technology.
3. The system model figure and simulation figures needed to be re-plotted. Especially, where are the simulation results in the simulation figures?
4.Some mistakes and error problems should be corrected in the revised manuscript.
Author Response
Reviewer 3
Comments and Suggestions for Authors
This paper studied a buffer-aided TDMA uplink network, where multiple status update devices and throughput demand devices are supposed to upload their data to one information access point (AP), and all devices are assumed to be provisioned with data buffer to temporarily store the randomly generated data from either the installed sensor or the upper-layer applications. This paper is interesting, however there are some problems that existed in this paper.
Reply to Comments and Suggestions for Authors:
The authors would like to thank the reviewer for the recognition of our work in this paper. We have tried to revise the manuscript by following the reviewer’s comments.
Comment 3-1:This paper investigated the TDMA topic, however, it is very old topic, why the authors investigated this topic the detailed reasons should be added into the revised manuscript
Reply to Comment 3-1:
The author greatly appreciates the reviewer's suggestions.
Basically, the TDMA is a mature and simple multiple access technology that has been widely applied and proven to be reliable in various communication environments. TDMA remains the preferred choice in communication system like satellite communication systems, emergency communication networks, and sensor networks. TDMA has shown its superiority due to its unique time allocation characteristics. In this case, TDMA can provide predictable latency and data freshness for every access device, just like the analysis in our manuscript. Of course, the use of advanced multiple access technique like NOMA and OFDMA can be employed to improve the AoI performance, studying how to effectively integrate our analysis into NOMA and OFDMA to further improve AoI performance will be left for our next step study.
Comment 3-2:When using the buffer technology, how to solve the delay problem when using this technology.
Reply to Comment 3-2:
The author would like to thank the reviewer for this comments.
Firstly, it should be addressed that, most of the current research on AoI focuses on the data generation model of generate at will [12] [19] [Wang 2019] [Akar 2021] [Liu 2021] [Kadota 2021]. However, this data generation model is too ideal and only focuses on the application layer to discuss the impact of scheduling decisions on AoI, hence unable to explore the impact of physical layer transmission capability on the realized AoI performance. In many IoT applications, such as the Internet of Vehicles, ecological environment temperature and humidity monitoring, and intelligent wearable devices for detecting human blood pressure, blood sugar, heartbeat, and other data, the generation of their data is more randomly generated based on real-time detection needs. Therefore, in our system model, the data generation model of the status update device considers a random arrival model. In addition, to ensure the integrity of the data and explore the impact of physical layer channel variations on the realized AoI performance of the device, we added data nuffer at the status update device to store the randomly arrived data packets of the upper layer application of the device, or the installed sensors.
Secondly, a large number of research results have confirmed us that the buffer-aided mechanism can be employed in order to improve the transmission rate of throughput demand devices. Just like the comments by the reviewer, the use of buffering strategy will incur queuing delay. Just like our analysis, in this case, the queuing delay will become a critical point to realize a reasonable trade-off between incurred increase in AoI for the status update devices and the improved transmission rate for the throughput demand devices. In fact, this is the most important contributions of our analysis. Our work provides useful guideline on how to devise the AoI-aware heterogenous multiple access scheme for practical IoT and sensor network.
[12] M. Moltafet, M. Leinonen, M. Codreanu and N. Pappas, "Power Minimization in Wireless Sensor Networks With Constrained AoI Using Stochastic Optimization," 2019 53rd Asilomar Conference on Signals, Systems, and Computers, Pacific Grove, CA, USA, 2019.
[19] J. Sun, L. Wang, Z. Jiang, S. Zhou and Z. Niu, "Age-Optimal Scheduling for Heterogeneous Traffic With Timely Throughput Constraints," IEEE Journal on Selected Areas in Communications, vol. 39, no. 5, pp. 1485-1498, 2021.
[Wang 2019] B. Wang, S. Feng and J. Yang, "When to preempt? Age of information minimization under link capacity constraint," in Journal of Communications and Networks, vol. 21, no. 3, pp. 220-232, June 2019.
[Akar 2021] N. Akar and O. Dogan, "Discrete-Time Queueing Model of Age of Information With Multiple Information Sources," in IEEE Internet of Things Journal, vol. 8, no. 19, pp. 14531-14542, 1 Oct.1, 2021.
[Liu 2021] L. Liu, H. H. Yang, C. Xu and F. Jiang, "On the Peak Age of Information in NOMA IoT Networks With Stochastic Arrivals," in IEEE Wireless Communications Letters, vol. 10, no. 12, pp. 2757-2761, Dec. 2021.
[Kadota 2021] I. Kadota and E. Modiano, "Age of Information in Random Access Networks with Stochastic Arrivals," IEEE INFOCOM 2021 - IEEE Conference on Computer Communications, Vancouver, BC, Canada, 2021.
Comment 3-3:The system model figure and simulation figures needed to be re-plotted. Especially, where are the simulation results in the simulation figures?
Reply to Comment 3-3:
The author greatly appreciates the reviewer's suggestions.
As per the reviewer’s comments, we have redrawn all the figures in the revised manuscript.
Comment 3-4:Some mistakes and error problems should be corrected in the revised manuscript.
Reply to Comment 3-4:
The author would like to thank the reviewer for the careful reading. We have tried our best to fix all typos in the revised manuscript.

Round 2
Reviewer 2 Report
Comments and Suggestions for Authors
The reviewer is satisfied with the revision.
Author Response
The authors would like the thank the reviewer for the recognition of our revision efforts.
Reviewer 3 Report
Comments and Suggestions for Authors
I have no more comments. This paper can be accepted in my opinion.
Author Response
The authors would like to thank the reviewer for the recognition of our revision efforts.